# Neuroprotective Action of Coumarin Derivatives through Activation of TRKB-CREB-BDNF Pathway and Reduction of Caspase Activity in Neuronal Cells Expressing Pro-Aggregated Tau Protein

**DOI:** 10.3390/ijms232112734

**Published:** 2022-10-22

**Authors:** Te-Hsien Lin, Kuo-Hsuan Chang, Ya-Jen Chiu, Zheng-Kui Weng, Ying-Chieh Sun, Wenwei Lin, Guey-Jen Lee-Chen, Chiung-Mei Chen

**Affiliations:** 1Department of Life Science, National Taiwan Normal University, Taipei 11677, Taiwan; 2Department of Neurology, Chang Gung Memorial Hospital, Chang Gung University College of Medicine, Taoyuan 33302, Taiwan; 3Department of Chemistry, National Taiwan Normal University, Taipei 11677, Taiwan

**Keywords:** tauopathies, TRKB signaling, pro-aggregated tau, neuroprotection, therapeutics

## Abstract

Hyperphosphorylation and aggregation of the microtubule binding protein tau is a neuropathological hallmark of Alzheimer’s disease/tauopathies. Tau neurotoxicity provokes alterations in brain-derived neurotrophic factor (BDNF)/tropomycin receptor kinase B (TRKB)/cAMP-response-element binding protein (CREB) signaling to contribute to neurodegeneration. Compounds activating TRKB may therefore provide beneficial effects in tauopathies. LM-031, a coumarin derivative, has demonstrated the potential to improve BDNF signaling in neuronal cells expressing pro-aggregated ΔK280 tau mutant. In this study, we investigated if LM-031 analogous compounds provide neuroprotection effects through interaction with TRKB in SH-SY5Y cells expressing ΔK280 tau_RD_-DsRed folding reporter. All four LMDS compounds reduced tau aggregation and reactive oxygen species. Among them, LMDS-1 and -2 reduced caspase-1, caspase-6 and caspase-3 activities and promoted neurite outgrowth, and the effect was significantly reversed by knockdown of TRKB. Treatment of ERK inhibitor U0126 or PI3K inhibitor wortmannin decreased p-CREB, BDNF and BCL2 in these cells, implying that the neuroprotective effects of LMDS-1/2 are via activating TRKB downstream ERK, PI3K-AKT and CREB signaling. Furthermore, LMDS-1/2 demonstrated their ability to quench the intrinsic fluorescence of tryptophan residues within the extracellular domain of TRKB, thereby consolidating their interaction with TRKB. Our results suggest that LMDS-1/2 exert neuroprotection through activating TRKB signaling, and shed light on their potential application in therapeutics of Alzheimer’s disease/tauopathies.

## 1. Introduction

Neurodegenerative tauopathies, including Alzheimer’s disease (AD), frontotemporal dementia, progressive supranuclear palsy and corticobasal syndrome, are characterized by accumulation of misfolded tau proteins [1]. These misfolded tau proteins are hyperphosphorylated and prone to aggregate into insoluble aggregations with a rich β-sheet structure [2]. It has been shown that tau aggregations increase oxidative stress and down-regulate the BDNF signaling pathway, eventually leading to neurodegeneration [3,4]. In addition, the misfolded tau protein propagates pathology through connected brain circuits in a prion-like manner [5]. As a result, identification of potential tau misfolding inhibitors is extremely needed to halt neurodegeneration of tauopathies.

Encoded by *MAPT*, tau plays an important role in axonal transport, microtubule dynamics and assembly [6,7,8]. Previously, a deletion mutation of *MAPT* within conserved 18-amino acid repeat domains of tau protein (∆K280 tau_RD_) was found in patients with tauopathies [9,10]. ∆K280 tau_RD_ demonstrates potential to accelerate tau misfolding [11]. Overexpression of ∆K280 tau_RD_ in SH-SY5Y cells induces overproduction of reactive oxygen species (ROS) and impairment of neurite outgrowth [12,13]. However, in neuronally differentiated SH-SY5Y cells, there is no significant difference in steady state levels of endogenous tau phosphorylation at Ser202, Thr231, Ser396 and Ser404 between this ΔK280 tau_RD_-expressed and un-expressed tau pro-aggregation cell model [14].

Brain-derived neurotrophic factor (BDNF) is a neurotrophic factor that promotes neuronal survival and growth [15]. Postmortem studies of AD patients demonstrate down-regulation of BDNF in hippocampus, cortex and basal nucleus of Meynert [16,17]. BDNF binds to tropomyosin-related kinase B (TRKB) to induce dimerization and phosphorylation of TRKB, and subsequent activation of downstream extracellular signal-regulated kinase (ERK) and phosphoinositide 3-kinase (PI3K)/protein kinase B (AKT) signaling pathways [18]. ERK phosphorylates cAMP responsive element binding protein 1 (CREB) to promote transcription of genes, such as BDNF [19] and B-cell lymphoma 2 (BCL2) anti-apoptosis regulator [20], for neuronal survival, neurite outgrowth and neuroplasticity [21,22]. In addition, the Pl3K-AKT pathway stimulates target gene expression via CREB phosphorylation to promote cell survival [23]. Enhancement of BDNF expression rescues neuronal death and improves learning and memory in rodent and primate models of AD [24,25,26]. However, the short plasma half-life and the limited penetration of the blood–brain barrier (BBB) restrict the potential application of BDNF [27]. Developing novel BBB-penetrating agonists of TRKB could overcome this limitation in treating neurodegenerative diseases.

Previously, we found that a novel coumarin derivative, LM-031, displayed neuroprotective potential by up-regulating nuclear factor erythroid 2-related factor 2 (NRF2) and CREB expression in pro-aggregatory tau SH-SY5Y cells and hyperglycemic triple-transgenic AD mice [28]. In this study, we evaluated the potential of LM-031 and its analogues LMDS-1 to -4 in treating tauopathies by examining if these compounds exert neuroprotective effects through enhancing TRKB signaling in ΔK280 tau_RD_-DsRed folding reporter SH-SY5Y cells. Their binding affinity to TRKB was also assessed by a tryptophan fluorescence quenching assay.

## 2. Results

### 2.1. Test Compounds

Coumarin derivates LM-031 and LMDS-1 to -4 were examined (Figure 1A). Anti-aggregation and anti-oxidative stress are important treatment approaches for neurodegenerative diseases. The inhibition of ∆K280 tau_RD_ aggregation was measured by thioflavin T assay. Congo red, known to reduce misfolded aggregation [29], was included for comparison. As shown in Figure 1B, EC_50_ values of Congo red, LM-031 and LMDS-1 to -4 for tau aggregation inhibition were: 10, 36, 84, 8, 21 and 14 μM, respectively. The free radical-scavenging activity was examined using DPPH as a substrate and kaempferol as a positive control [30]. Kaempferol, LM-031 and LMDS-1 to -4 had EC_50_ values of 28, 93, 122, 132, 132 and 126 μM, respectively (Figure 1C). In addition, the oxygen radical absorbance capacity of the LM-031 and LMDS-1 to -4 was examined based on a Trolox standard curve. LM-031 and LMDS-1 to -4 at 100 μM had activity equivalent to 51, 10, 15, 12 and 16 μM Trolox, respectively (Figure 1D). The cytotoxicity of LM-031 and LMDS-1 to -4 was examined by MTT assay. All compounds had cell viability up to 75–92% in 100 µM compound-treated ∆K280 tau_RD_-DsRed SH-SY5Y cells (Figure 1E).

### 2.2. Inhibition of ΔK280 Tau_RD_ Aggregation and Oxidative Stress in ∆K280 Tau_RD_-DsRed SH-SY5Y Cells

∆K280 tau_RD_-DsRed SH-SY5Y folding reporter cells [12] were used to evaluate cellular ∆K280 tau_RD_ aggregation inhibition (Figure 2A). The misfolded ΔK280 tau_RD_ formed aggregates to adversely affect the proper folding of fused DsRed and, thus, decrease DsRed fluorescence [13]. Congo red, known to attenuate aggregated tau-induced toxicity [31], was included for comparison. Treatment with Congo red, LM-031 or LMDS-1 to -4 at 10 µM concentration significantly increased the DsRed fluorescence intensity (108–113%, *p* = 0.047–0.013; cell viability: 88–80%) (Figure 2B), without affecting ∆K280 tau_RD_-DsRed RNA level (24.7–25.4 folds, *p* > 0.05) (Figure 2C). Based on the cell number analyzed, the IC_50_ values of Congo red, LM-031, LMDS-1, LMDS-2, LMDS-3 and LMDS-4 were 45, 43, 25, 24, 29 and 25 μM, respectively (Figure 2B). In oxidative stress analysis, ∆K280 tau_RD_-DsRed expression elevated the ROS level of ∆K280 tau_RD_-DsRed-expressing SH-SY5Y cells (115%, *p* = 0.015), while treatments with Congo red, LM-031 and LMDS-1 to -4 at 10 μM concentration effectively reduced the ROS level associated with ∆K280 tau_RD_ overexpression (102–90%, *p* = 0.042—<0.001) (Figure 2D). These results suggested that LM-031 and LMDS-1 to -4 not only inhibited ∆K280 tau_RD_ aggregation, but also attenuated oxidative stress induced by ∆K280 tau_RD_ overexpression.

### 2.3. Neuroprotective Effects of LM-031 and LMDS-1 to -4

ROS overproduction induces brain inflammation via caspase-1 activation, and the stress subsequently induces caspase-6 activation to lead to axonal degeneration in AD [32]. Inhibition of caspase-1 alleviates neuropathology and improves cognitive deficits in mice with two familial AD mutations (APP KM670/671NL and V717F) [33]. Caspase-6 knockout in mice with five familial AD mutations (APP KM670/671NL, I716V and V717I, PSEN1 M146L and L286V) also reveals favorable outcome on memory and neurological hallmarks [34]. Therefore, the neuroprotective effects of LM-031 and LMDS-1 to -4 including neurite outgrowth as well as caspase-1 and caspase-6 activities were evaluated. The overexpression of ∆K280 tau_RD_ significantly reduced neurite length (from 30.9 μm to 27.2 μm, *p* = 0.013) and branching (from 0.99 to 0.87, *p* = 0.009). Treatment with Congo red, LM-031, LMDS-1 or -2 (10 µM) successfully rescued the impairment of neurite length (from 27.2 μm to 30.3–31.5 μm, *p* = 0.045–0.003) and branching (from 0.87 to 1.00–1.06, *p* = 0.005 – <0.001) (Figure 3A). In addition, the overexpression of ∆K280 tau_RD_ raised caspase-1 activity (126%, *p* = 0.065), and treatment with Congo red, LM-031, LMDS-1 and -2 (10 µM) reduced the caspase-1 activity compared to no treatment (from 126% to 85–75%; *p* = 0.002 – <0.001) (Figure 3B). Treatment with Congo red, LM-031, LMDS-1/2 (10 µM) also reduced the caspase-6 (from 111% to 96–94%; *p* = 0.004–0.001) (Figure 3C) and caspase-3 activity (from 113% to 100–98%; *p* = 0.043–0.021) (Figure 3D) compared to no treatment.

### 2.4. TRKB Expression and Knockdown in ∆K280 Tau_RD_-DsRed SH-SY5Y Cells

Since LMDS-1 to -4 showed neuroprotective effects on ∆K280 tau_RD_-DsRed SH-SY5Y cells, we knocked down TRKB expression through lentivirus-mediated shRNA targeting in these cells to evaluate the potential of TRKB and downstream signaling as therapeutic targets of LM-031 and LMDS-1 to -4 (Figure 4A). In scrambled shRNA-infected cells, ∆K280 tau_RD_-DsRed overexpression did not significantly affect TRKB expression (95%; *p* > 0.05). Treatment with LM-031 and LMDS-1 to -4 also did not affect TRKB expression (86–102%; *p* > 0.05). However, TRKB-specific shRNA reduced TRKB level in ∆K280 tau_RD_-DsRed cells (from 95% to 27%, *p* < 0.001) (Figure 4B). TRKB-specific shRNA also reduced TRKB level in these cells treated with LM-031 and LMDS-1 to -4 (from 86–102% to 23–27%; *p* < 0.001) (Figure 4B).

The neurite outgrowth-promoting effects of LM-031 and LMDS-1 to -4 were also evaluated in the above TRKB-knockdown cells. ∆K280 tau_RD_-DsRed overexpression significantly reduced the neurite length (from 31.3 µm to 26.9 µm) and branching (from 1.07 to 0.82) (*p* < 0.001) (Figure 4C). TRKB-specific shRNA further reduced neurite length/branching to 24.2 µm/0.65 (*p* = 0.030–0.029) in these cells (Figure 4C). In ∆K280 tau_RD_-DsRed cells, treatment with LM-031, LMDS-1/2 rescued the reduced neurite length (from 24.2 µm to 29.6–30.0 µm, *p* = 0.040–0.009) and branching (from 0.65 to 1.05–1.07, *p* < 0.001), and the rescue was counteracted by TRKB-specific shRNA (length: 26.5–24.7 µm, *p* = 0.011 – <0.001; branching: 0.69–0.63; *p* < 0.001) (Figure 4C). Therefore, LMDS-1/2 were selected for further investigations.

### 2.5. Therapeutic Targets of LMDS-1/2 in ∆K280 Tau_RD_-DsRed SH-SY5Y Cells

The effects of LMDS-1/2 on expression levels of ERK, AKT and downstream targets of TRKB were examined by applying ERK inhibitor U0126 or PI3K inhibitor wortmannin (10 μM) to LMDS-1/2-treated SH-SY5Y cells expressing ∆K280 tau_RD_-DsRed (Figure 5A). Overexpression of ∆K280 tau_RD_-DsRed reduced p-ERK (76%, *p* = 0.066), and LMDS-1/2 treatment raised p-ERK (101–102%, *p* = 0.048–0.038). A similar trend of reduced p-AKT with ∆K280 tau_RD_-DsRed overexpression (72%, *p* = 0.012) and elevated p-AKT with LMDS-1/2 treatment (96–100%, *p* = 0.042–0.012) was also observed. U0126 treatment mitigated the increase in p-ERK (from 101–102% to 65–66%, *p* = 0.003–0.002) and wortmannin treatment attenuated the up-regulation of p-AKT (from 96–100% to 62–64%, *p* = 0.004 – <0.001) (Figure 5B). Moreover, induced expression of ∆K280 tau_RD_-DsRed reduced p-TRKB Y516 (67%, *p* = 0.006) and Y817 (74%, *p* = 0.042), p-CREB (78%, *p* = 0.020), CREB (83%, *p* = 0.023), pro- (62%, *p* = 0.021) and m-BDNF (69%, *p* = 0.012) and BCL2 (73%, *p* = 0.043), and increased BAX (131%, *p* = 0.040), and treatment with LMDS-1/2 increased p-TRKB Y516 (92–93%, *p* = 0.046–0.036) and Y817 (103–105%, *p* = 0.020–0.012), p-CREB (105–108%, *p* = 0.004–0.001), CREB (98–101%, *p* = 0.048–0.013), pro- (94–99%, *p* = 0.071–0.027) and m-BDNF (97–102%, *p* = 0.024–0.007) and BCL2 (111–114%, *p* = 0.003–0.001), and reduced BAX (94–90%, *p* = 0.011–0.004) (Figure 5C). Notably, treatment with U0126 or wortmannin attenuated the increase in p-CREB (81–75%, *p* = 0.007–0.002) and CREB (80–77%, *p* = 0.013–0.001), and BCL2 (88–83%, *p* = 0.043–0.015), while the increase in pro- and m-BDNF or decrease in BAX was not affected by U0126 or wortmannin. Unexpectedly, treatment with U0126 attenuated the increase in p-TRKB Y516 (65–57%, *p* = 0.022–0.003) and Y817 (69–67%, *p* = 0.006–0.002) by LMDS-1/2.

### 2.6. Comparative Effects of LMDS-1/2 and BDNF on TRKB and CREB Expression

As LMDS-1/2 target TRKB and downstream CREB, effects of BDNF and LMDS compounds on TRKB and CREB expression were compared. Inhibition of ΔK280 tau_RD_ aggregation and oxidative stress were first evaluated by treating ∆K280 tau_RD_-DsRed SH-SY5Y cells with BDNF at 10–100 ng/mL concentration or LMDS-1/2 at 5–10 μM concentration (Figure 6A). BDNF at 100 ng/mL and LMDS-1/2 at 10 μM effectively increased the DsRed fluorescence intensity (111–112%, *p* = 0.003–0.001) and reduced the ROS induced by ∆K280 tau_RD_-DsRed overexpression (from 115% to 98–88%, *p* = 0.012—<0.001). BDNF at 100 ng/mL and LMDS-1/2 at 10 μM were then selected to compare the efficacy in TRKB and CREB expression (Figure 6B). Overexpression of ∆K280 tau_RD_-DsRed down-regulated p-TRKB (77–83%, *p* = 0.011–0.010) and p-CREB (72%, *p* = 0.003), whereas BDNF, LMDS-1/2 treatment rescued the reduction (p-TRKB: 96–104%, *p* = 0.049–0.003; p-CREB: 90–95%, *p* = 0.044–0.012). No significant differences in p-TRKB and p-CREB were detected between BDNF and LMDS-1/2 groups (*p* > 0.05).

In addition to activating the TRKB receptor, BDNF also binds to low-affinity nerve growth factor receptor (NGFR) [35], a member of the tumor necrosis factor receptor superfamily. NGFR was expressed in the membrane fractions of SH-SY5Y cells [36]. Increased expression of the NGFR in plasma membrane enhanced mitogen activated protein kinase 8 (JNK) activation and apoptotic cell death in SH-SY5Y cells [37]. We thus examined if these LMDS compounds may act on NGFR. As shown in Figure 6B, LMDS-1/2 treatments did not increase p-JNK (T183/Y185) expression in SH-SY5Y cells expressing ∆K280 tau_RD_-DsRed (93–108% versus 107%, *p* > 0.05), suggesting no enhancement of JNK activation through NGFR binding.

### 2.7. Evaluation of TRKB Binding Affinity

The induction of p-TRKB indicated the interaction between LMDS-1/2 and TRKB. Therefore, we cloned into *Pichia* expression vector pGAPZα A (Figure 7A) for constitutive expression and purification of recombinant TRKB-ECD-His protein (Figure 7B). The prepared His-tagged TRKB-ECD protein was used to determine binding specificity of LM-031, LMDS-1 and LMDS-2 via tryptophan fluorescence quenching assay. 7,8-DHF, a selective TRKB agonist with potent neurotrophic activities [38], was included for comparison. The assay relies on the ability to quench the intrinsic protein fluorescence of tryptophan residues which can be selectively measured by exciting at 295 nm. The tryptophan fluorescence differences of TRKB-ECD-His in the presence of test compounds (1–1000 nM) were examined, as binding of 7,8-DHF to TRKB receptor purified from Chinese hamster ovary cells induced a change of the microenvironment of the tryptophan, which was captured by fluorescence spectroscopy [39]. As shown in Figure 7C, tryptophan fluorescence of TRKB-ECD-His was quenched by 7,8-DHF, LM-031, LMDS-1 or LMDS-2 in a concentration-dependent manner, and the fluorescence decrease was maximal at the highest concentration of test compounds. Based on the quantitative analysis of fluorescence change, TRKB-ECD binding affinity (*K_D_*) of 7,8-DHF was 16.0 ± 3.4 nM (Figure 7D). The observed *K_D_* of 12.7 ± 2.8 nM, 8.0 ± 17.0 nM and 6.5 ± 6.6 nM of LM-031, LMDS-1 and LMDS-2 demonstrated the high binding affinities to the ECD of the TRKB receptor.

## 3. Discussion

The pathogenesis of tauopathies is still unclear, while the treatment to halt tau-mediated neurodegeneration remains unavailable. Here, we demonstrate that overexpression of pro-aggregated ∆K280 tau_RD_ in SH-SY5Ycells leads to increased aggregation and oxidative stress, neurite outgrowth defects, and up-regulation of caspase-1 and caspase-6 activities. ∆K280 tau_RD_ also down-regulates the phosphorylation of TRKB and CREB, as well as the expression of CREB, BDNF and BCL2, and increases BAX expression. Administrations with coumarin derivatives LMDS-1/2 rescue these neurodegenerative phenotypes by increasing TRKB phosphorylation and BDNF expression. Knockdown of TRKB and treatments with ERK inhibitor U0126 or PI3K inhibitor wortmannin counteract the neuroprotective effects of LMDS-1/2. The tryptophan fluorescence quenching assay further confirms the direct interactions between LMDS-1/2 and the TRKB extracellular domain. These results shed light on the role of TRKB signaling in tau-mediated neurodegeneration. The neuroprotective effects of LMDS-1/2 further suggest the potential of TRKB agonists in treating tauopathies.

A few TRKB agonists have been examined in cell or animal models for neurodegenerative diseases. For example, 7,8-DHF has the potential to attenuate Aβ deposition, loss of hippocampal synapses and memory deficits in transgenic AD mice [40,41]. 7,8-DHF also reduces ROS production in HT-22 hippocampal neuronal cells [42], and protects PC12 pheochromocytoma cells against 6-hydroxydopa-induced cell death [43]. Intranasal administration with LM22A-4 activates TRKB to improve motor learning after traumatic brain injury in rats [44]. However, the limited ability for crossing the BBB restricts the application of LM22A-4 in treating neurodegenerative diseases [44]. In our study, LMDS-1/2 directly bind to TRKB, elicit downstream signaling, including ERK and PI3K-AKT pathways, and protect neurons against tau-mediated neurotoxicity. In the tryptophan fluorescence quenching assay, LMDS-1/2 demonstrate adequate binding affinity to TRKB. Both compounds reduce ROS and improve neurite outgrowth in ΔK280 tau_RD_-DsRed SH-SY5Y cells. Blockage of either ERK or Pl3K-AKT signaling counteracts the improvement of neurite outgrowth by LMDS-1/2, suggesting activations of both signals are necessary for neuroprotection.

Highly expressed in neural tissues, ERK plays an important role in the survival of neurons [45]. Transgenic activation of ERK in mice reduces the 1-methyl-4-phenyl-1,2,3,6-tetrahydropyridine (MPTP)- or facial nerve axotomy-induced loss of dopaminergic or motor neurons [46]. On the other hand, a number of reports have shown a relationship between neurodegeneration and increased ERK activity. Activation of ERK has been found in hippocampal slides from *APP*-transgenic AD mice [47]. ERK is colocalized with tau and neurofibrillary tangles in neurons [48]. The phosphorylation of ERK is elevated in brain extracts from AD patients [49]. Tau is also a substrate of phosphorylation by ERK [50]. Importantly, ERK signaling can be activated by c-reactive protein or complements [51,52], both of which are colocalized with neurofibrillary tangles in brains of AD patients [52,53,54]. The complex role of ERK activation and its interactions with other signaling pathways in neurodegeneration need to be elucidated.

The PI3K-AKT signaling pathway plays an important role in neuronal survival, proliferation, differentiation, autophagy and neurogenesis [55]. The activation of the PI3K-AKT signal pathway is able to protect neurons against tau aggregation. In AD, the activation of glycogen synthase kinase-3β (GSK-3β) is closely related to the hyperphosphorylation and accumulation of tau [56]. GSK-3β is rendered inactive when it is phosphorylated at S9 by AKT [57]. The postmortem study showed decreased levels of PI3K subunits and reduced phosphorylation of AKT in the brains of AD patients [58,59]. Administering AKT inhibitor wortmannin to rat lateral ventricles increases tau hyper-phosphorylation and causes axonal swelling [60]. In our study, inhibition of AKT by wortmannin attenuates the rescue of neurite outgrowth by LMDS-1/2, suggesting that the PI3K-AKT signaling pathway is mandatory for axon growth. On the other hand, tau is also a phosphorylation substrate of AKT, while AKT inhibition leads to the decrease in tau phosphorylation at T212 and S214 [61,62]. Importantly, TRKB-PI3K-AKT signaling also promotes the activation of NRF2 antioxidant signaling [63], while disruption of antioxidant signaling increases the production of ROS and further impairs cognitive function in animal models [64,65,66,67]. The net effect of PI3K-AKT inhibition in tauopathies should be further investigated.

The activations of ERK and PI3K-AKT by LMDS-1/2 further phosphorylate CREB. CREB is a critical nuclear transcription factor needed for neuron survival [22]. Phosphorylated CREB binds to the coactivators CREB binding protein (CBP) and E1A binding protein p300 (EP300) to up-regulate the expression of target genes such as BDNF and BCL2 [20,68]. It is of interest that the promoter of tau gene *MAPT* contains CRE-like elements and overexpression of CREB down-regulates the expression of tau in SH-SY5Y cells [69]. Our results show that application of either U0126 or wortmannin down-regulates CREB phosphorylation and expression of BDNF and BCL2, and up-regulates BAX expression, supporting the regulation of CREB activity by ERK and PI3K-AKT signaling pathways. The neuroprotection of LMDS-1/2 against tau-mediated neurotoxicity is also counteracted by the treatment of both inhibitors, implicating the contribution of ERK, PI3K-AKT and CREB signaling pathways to the neuroprotection in the ΔK280 tau_RD_ cell model. Furthermore, our group also demonstrated that LMDS-1 could up-regulate the TRKB-ERK-CREB pathway and BDNF expression in hippocampal primary neurons incubated with oligomeric Aβ and Aβ-induced AD mice [70]. These results support the potential of coumarin derivatives in improving neurodegeneration.

The fluorescent tryptophan quenching assay supports that LMDS-1 and LMDS-2 directly interact with TRKB-ECD. The *K_D_* of these two compounds are lower compared to 7,8-DHF and LM-031, suggesting their high affinities to TRKB-ECD, whereas their binding sites in TRKB-ECD remain elusive. BDNF interacts with the leucine-rich repeat (LRR) motif and Ig2 domain of TRKB-ECD [71]. Molecular modeling supports that 7,8-DHF could insert itself into to the pocket between the N-terminal cap and the first repeat of the LRR domain [72]. However, TRKB-ECD is a highly glycosylated protein that contains 10 N-linked glycosylation sites [73]. Six disulfide linkages are also formed within two cysteine cluster domains in the N-terminus of TRKB [73]. It remains uncertain whether *Pichia pastoris*-derived recombinant proteins could recapitulate these posttranslational modifications. Further studies, such as the cocrystal structure analysis, would provide more spatial details on the direct interaction between LMDS1/2 and TRKB. Apart from binding to TRKB, LMDS-1/2 also directly reduce the aggregation of *E. coli*-derived ΔK280 tau_RD_ in thioflavin T assay. It is also possible that the reduction of tau aggregation can modulate ERK, PI3K-AKT and CREB signaling pathways. Alternatively, LMDS-1/2 may also activate other receptors to modulate the above signaling pathways and their neuroprotective effects. Further studies will be warranted to clarify whether ERK, PI3K-AKT and CREB are up-regulated directly by TRKB activation, by aggregate inhibition, or by a pleiotropic mechanism through activation of other receptors. In vivo experimentation is also needed to confirm these promising results.

## 4. Materials and Methods

### 4.1. Coumarin Compounds

Coumarin derivatives LMDS-1 to -4 were purchased from Enamine (Kyiv, Ukraine). In-house LM-031 activating the CREB-dependent survival and anti-apoptosis pathway [26] was included for comparison. In addition, Congo red and kaempferol were obtained from Sigma-Aldrich Co. (St. Louis, MO, USA) as positive controls for monitoring tau folding and free radical-scavenging activity, respectively.

### 4.2. Anti-ΔK280 Tau_RD_ Aggregation Assay

Thioflavin T is widely used to visualize and quantify the presence of misfolded amyloid protein aggregates in vitro [74]. The tau aggregation inhibiting potential of Congo red, LM-031 and LMDS-1 to -4 was assessed by using *E. coli*-expressed ΔK280 tau_RD_ protein [28]. Briefly, purified ΔK280 tau_RD_ protein (20 μM) was incubated with Congo red or coumarin compounds (1–10 μM) in 150 mM NaCl and 20 mM Tris-HCl pH8.0, at 37 °C with shaking for 48 h to form aggregates. Then, thioflavin T (final 5 μM; Acros Organics, Geel, Belgium) was added and incubated for 25 min at room temperature. Fluorescence intensity of samples was recorded at excitation/emission wavelengths of 420/485 nm using an FLx800 microplate reader (BioTek, Winooski, VT, USA). The half maximal effective concentration (EC_50_) values were calculated by using a linear interpolation method.

### 4.3. Antioxidant Assay

The potential free radical-scavenging activity of test compounds was assayed using stable free radical 1,1-diphenyl-2-picrylhydrazyl (DPPH) (Sigma-Aldrich). Briefly, kaempferol, LM-031 or LMDS-1 to -4 (10–80 µM) was added to DPPH (100 µM in 99% ethanol). The solution was vortexed for 15 s, left for 30 min at room temperature, the mixture was transferred to a 96-well UV-transparent microplate and the absorbance at 517 nm was measured using a Multiskan^TM^ GO microplate spectrophotometer (Thermo Fisher Scientific, Waltham, MA, USA). Radical-scavenging activity was calculated using the equation: 1 − (absorbance of sample/absorbance of control) × 100%, and EC_50_ estimated using the linear interpolation method.

In addition, an oxygen radical antioxidant capacity (ORAC) assay was performed using an OxiSelect™ kit according to the manufacturer’s instruction (Cell Biolabs, San Diego, CA, USA). Briefly, Trolox standards (2.5–50 μM) and samples (4–100 μM) were diluted with 50% acetone. Blank (50% acetone), standards and samples were mixed with fluorescein and incubated at 37 °C for 30 min. Following the free radical initiator 2,2′-azobis(2-methylpropionamidine) dihydrochloride (AAPH), the produced peroxyl radicals (ROO•) quench the fluorescent probe over time [75]. Antioxidants present in the assay block the peroxyl radical oxidation of the fluorescent probe. The course of the reaction was recorded for 60 min, with one measurement every five minutes using a BioTek FLx800 microplate reader. The excitation and emission wavelengths were set at 480 nm and 520 nm, respectively. To quantify the oxygen radical antioxidant activity in a sample, the area under the curve (AUC) for blank, standard and samples was calculated. After subtraction of the blank, the equivalent Trolox concentrations of samples were expressed based on the Trolox standard curve.

### 4.4. Cells and Culture

Tet-On neuroblastoma SH-SY5Y-derived ∆K280 tau_RD_-DsRed cells [12,13] were maintained in Dulbecco’s modified Eagle medium/nutrient mixture F12 (DMEM/F12) supplemented with 10% fetal bovine serum (FBS) (Thermo Fisher Scientific, Waltham, MA, USA) at 37 °C under 5% CO_2_ and 95% relative humidity, with 5 µg/mL blasticidin and 100 µg/mL hygromycin (InvivoGen, San Diego, CA, USA) added to the growth medium. Expression of ∆K280 tau_RD_-DsRed can be induced following addition of doxycycline (2 μg/mL; Sigma-Aldrich).

### 4.5. Cytotoxicity Assay

To evaluate cytotoxicity of each compound, ∆K280 tau_RD_-DsRed SH-SY5Y cells were dispensed at 3 × 10^4^ cells/100 μL in 96-well dishes, grown for 20 h and treated with the test compounds (0.1–100 μM). On the next day, 10 μL of 5 mg/mL tetrazolium dye 3-(4,5-dimethylthiazol-2-yl)-2,5-diphenyltetrazolium bromide (MTT) (Sigma-Aldrich) was added to the cells at 37 °C for 3 h, followed by addition of 100 μL lysis buffer (10% Triton X-100, 0.1 N HCl, 18% isopropanol) for 16 h to dissolve the insoluble formazan crystals, which has a purple color. The absorbance of this colored solution was quantified by measuring at 570 nm by using a Multiskan^TM^ GO microplate spectrophotometer (Thermo Fisher).

### 4.6. High-Content Analysis of ∆K280 Tau_RD_-DsRed Fluorescence and Oxidative Stress

On day 1, ∆K280 tau_RD_-DsRed SH-SY5Y cells were seeded in a 96-well plate (3 × 10^4^/well), with retinoic acid (10 µM; Sigma-Aldrich) added to induce neuronal differentiation [76]. On day 2, cells were pretreated with BDNF (10–100 ng/mL), Congo red, LM-031 or LMDS-1 to -4 (2.5–10 µM) for 8 h, followed by inducing ∆K280 tau_RD_-DsRed expression with doxycycline (2 μg/mL). On day 8, cells were stained with Hoechst 33342 (0.1 µg/mL; Sigma-Aldrich) for 30 min and cell images were automatically recorded at excitation/emission wavelengths of 543/593 nm (ImageXpress Micro Confocal High-Content Imaging System) and analyzed (MetaXpress Microscopy Automation and Image Analysis Software version 6) (Molecular Devices, Sunnyvale, CA, USA). For ROS measurement, dichloro-dihydro-fluorescein diacetate (DCFH-DA, 10 µM; Invitrogen, Carlsbad, CA, USA) was added to the cells on day 8 and incubated at 37 °C for 30 min. ROS in cells was measured using the high-content analysis system, with excitation/emission wavelengths at 482/536 nm.

### 4.7. Real-Time PCR Analysis

Total RNA of ∆K280 tau_RD_-DsRed SH-SY5Y cells on day 8 was extracted using Trizol reagent (Sigma-Aldrich) and reverse transcribed using SuperScript^TM^ III reverse transcriptase (Invitrogen). Quantitative PCR was performed with 50 ng cDNA and customized Assays-by-Design probe for DsRed [12] and HPRT1 (4326321E, endogenous control) in a 96-well real-time PCR instrument (StepOnePlus^TM^ Real-time PCR system; Applied Biosystems, Foster City, CA, USA). Fold difference of RNA was calculated using the formula 2^∆Ct^, ∆C_T_ = C_T_ (HPRT1) − C_T_ (DsRed), in which C_T_ indicates cycle threshold.

### 4.8. High-Content Analysis of Neurite Outgrowth

As described, ∆K280 tau_RD_-DsRed SH-SY5Y cells were seeded in a 24-well plate (5 × 10^4^/well) with retinoic acid addition on day 1, treated with tested compounds (10 µM) and ∆K280 tau_RD_-DsRed expression was induced with doxycycline (2 μg/mL) on day 2. On day 8, after being fixed in 4% paraformaldehyde for 15 min, permeabilized in 0.1% Triton X-100 for 10 min and blocked in 3% bovine serum albumin (BSA) for 20 min, the cells were stained with TUBB3 (neuronal class III β-tubulin) primary antibody (1:1000; BioLegend, San Diego, CA, USA) at 4 °C overnight, followed by goat anti-rabbit Alexa Fluor^®^ 555 secondary antibody (1:1000; Molecular probes) at room temperature for 2 h, with 4′-6-diamidino-2-phenylindole (DAPI, 0.1 µg/mL; Sigma-Aldrich) included for nuclei staining. Neuronal images were captured using the high-content analysis system as described. Neurite total length (μm), processes (primary neurite extensions projecting directly from the cell body) and branching (points at which primary neurites bifurcated) were analyzed using Neurite Outgrowth Application Module (MetaXpress; Molecular Devices). Around 5000 cells were analyzed in each of three independent experiments for each sample.

### 4.9. Caspase-1, -3 and -6 Activity Assays

∆K280 tau_RD_-DsRed SH-SY5Y cells were seeded in a 6-well plate (5 × 10^5^/well) and treated with retinoic acid, test compound and doxycycline as described. On day 8, cells were collected and resuspended in lysis buffer, followed by six freeze/thaw cycles and centrifugation to collect the supernatant. Caspase-1 activity in cell extracts was measured using a caspase-1 fluorometric assay kit (YVAD-AFC substrate; BioVision, Milpitas, CA, USA). In addition, a caspase-6 fluorometric assay kit (VEID-AFC substrate; BioVision) was used to measure caspase-6 activity. The mixture was incubated for 2 h at 37 °C. A FLx800 microplate reader (BioTek) was used to measure caspase-1 or caspase-6 activity with excitation and emission wavelengths of 400 nm and 505 nm, respectively. For caspase-3 activity measurement, a caspase-3 fluorometric assay kit (DEVD-AMC substrate, Sigma-Aldrich) was used. The excitation and emission wavelengths of AMC were 360 nm and 460 nm, respectively.

### 4.10. RNA Interference

Lentiviral short hairpin RNA (shRNA) targeting TRKB (TRCN0000002243, TRCN0000002245 and TRCN0000002246) and a negative control scrambled (TRC2.Void, ASN000001) [77] from National RNAi Core Facility, IMB/GRC, Academia Sinica, Taipei, Taiwan were applied to knock down TRKB expression in ∆K280 tau_RD_-DsRed SH-SY5Y cells. On day 1, cells were plated on 6-well (for protein analysis) or 24-well (for neurite outgrowth analysis) plates in the presence of retinoic acid as described. On day 2, the cells were infected with lentivirus (multiplicity of infection of 3 for each shRNA) in medium containing polybrene (8 µg/mL; Sigma-Aldrich) to increase the efficiency of viral infection. On day 3, the culture medium was changed and the cells were pretreated with LM-031 or LMDS-1 to -4 (10 µM) for 8 h, followed by inducing ∆K280 tau_RD_-DsRed expression. On day 9, the cells were collected for TRKB protein analysis or analyzed for neurite outgrowth as described.

### 4.11. Kinase Inhibitor Treatment

∆K280 tau_RD_-DsRed SH-SY5Y cells in 6-well plates (5 × 10^5^/well) were treated with retinoic acid (10 µM) on day 1, followed by LMDS-1 or -2 (10 µM) and doxycycline (2 µg/mL) addition on day 2 as described. ERK inhibitor U0126 or PI3K inhibitor wortmannin (10 μM) (LC Laboratories, Woburn, MA, USA) was added on day 6. On day 8, the cells were collected for BDNF, BCL2, BAX and total/phosphorylated TRKB, ERK, AKT, CREB protein analyses as described.

### 4.12. Protein Blot Analysis

∆K280 tau_RD_-DsRed SH-SY5Y cells were lysed in buffer containing 50 mM Tris-HCl (pH 8.0), 150 mM NaCl, 1 mM EDTA (pH 8.0), 1 mM EGTA (pH 8.0), 0.1% SDS, 0.5% sodium deoxycholate, 1% Triton X-100 and protease (Sigma-Aldrich)/phosphatase (Roche, Basel, Switzerland) inhibitor cocktails. After two sonications with an interval of 30 min on ice, cell lysates were collected by centrifugation at 14,000× *g* for 30 min at 4 °C. Protein concentrations were determined using a Bradford assay (Bio-Rad, Hercules, CA, USA). Protein samples (25 µg) were separated by 10–12% SDS-PAGE and transferred to polyvinylidene difluoride (PVDF) membranes (Sigma-Aldrich). After blocking for 2 h with 10% skimmed milk or 5% BSA in 0.1% Tween 20 in Tris-buffered saline (TBST), the membrane was incubated with primary antibody against BDNF (1:500; Santa Cruz, Santa Cruz, CA, USA), BCL2 (1:500; BioVision), BAX (1:500; BioVision), TRKB (1:1000; Cell Signaling, Danvers, MA, USA), p-TRKB (Y516) (1:1000; Cell Signaling), p-TRKB (Y817) (1:1000; Millipore, Billerica, MA, USA), ERK (1:1000; Cell Signaling), p-ERK (T202/Y204) (1:500; Cell Signaling), AKT (1:1000; Cell Signaling), p-AKT (S473) (1:1000; Cell Signaling), CREB (1:1000; Santa Cruz), p-CREB (S133) (1:1000; Millipore), JNK (1:1000; Cell Signaling), p-JNK (T183/Y185) (1:1000; Cell Signaling) or loading control glyceraldehyde-3-phosphate dehydrogenase (GAPDH) (1:1000; MDBio, Taipei, Taiwan). After three washes with TBST, the immune complexes were detected using horseradish peroxidase-conjugated goat anti-mouse or goat anti-rabbit secondary IgG antibody (1:5000, GeneTex, Irvine, CA, USA) and enhanced chemiluminescence substrates luminol and peroxide (Millipore). The primary and secondary antibodies used were diluted with TBST. The chemiluminescent detection was performed on ImageQuant™ LAS 4000mini (GE Healthcare).

### 4.13. TRKB-ECD Protein Preparation

The cDNA fragment containing the full-length extracellular domain of TRKB (TRKB-ECD) (residues 32–429, XP_016870240) was amplified from pDONR223-NTRK2 plasmid (Addgene, Watertown, MA, USA) using forward primer 5′-ctcgagaaaagagaggctgaagctTGTCCCACGTCCTGAAATG (*Xho*I site and Kex2 cleavage of the signal sequence in lowercase letters) and reverse primer 5′-gtcgacTTCCCGACCGGTTTTATCAGTG (*Sal*I site in lowercase letters). After cloning and sequencing, the TRKB-ECD fragment was excised with *Xho*I and *Sal*I and subcloned into *Pichia* expression vector pGAPZα A (Invitrogen). The *Avr*II-linearized plasmid was then transformed into *Pichia pastoris*, and stable transformants selected and cultured at 30 °C for 4 days for constitutive expression of recombinant His-tagged TRKB-ECD. The secreted TRKB-ECD-His in cultured medium was purified using Ni-NTA His•Bind resins (Novagen, Madison, WI 53719, USA) according to the supplier’s instructions. Protein purity was examined by Coomassie blue staining of SDS-polyacrylamide gel as well as immunoblotting using anti-His (1:2000; Acris, Herford, Germany) and anti-TRKB (1:1000; Cell Signaling) antibodies.

### 4.14. Tryptophan Fluorescence Quenching Assay

The interaction between TRKB-ECD-His and LM-031, LMDS-1 or LMDS-2 was determined by tryptophan fluorescence titration. 7,8-DHF (Sigma-Aldrich) was included as a positive control [39]. Tryptophan fluorescence spectra of TRKB-ECD-His (50 nM in PBS) were obtained before and after titration with different concentrations of test compounds (0–1000 nM). The fluorescent quenching was conducted in a cuvette with a 1 cm path length cell by an F-7000 fluorescence spectrophotometer (Hitachi, Tokyo, Japan) at room temperature. The intrinsic fluorescence spectra of TRKB were recorded with excitation wavelength set at 295 nm and emission wavelength in the range 300–400 nm. The dissociation constant (*K_D_*) of 7,8-DHF, LM-031, LMDS-1 and LMDS-2 to TRKB was determined by fitting the normalized fluorescence intensity at 330 nm under different concentrations of test compounds using the equation derived from Liu et al. [39]:*K_D_* = [*P*]*_T_* (*f* − 1) + [*D*]*_T_* (1/*f* − 1),
where *f* is the fractional change, and [*P*]*_T_*and [*D*]*_T_*are the total concentration of TRKB-ECD-His protein and test compound, respectively.

### 4.15. Statistical Analysis

The presented data are shown as mean ± SD of three independent experiments in two or three biological replicates. To compare the differences between groups, two-tailed Student’s *t*-test or one-way analysis of variance (ANOVA) was performed, with Tukey’s post hoc test where appropriate. *p* values < 0.05 were considered as statistically significant.

## 5. Conclusions

In the present study, we show that the TRKB signaling pathway in ΔK280 tau_RD_ SH-SY5Y cells is compromised, resulting in BCL2 and BDNF down-regulations, BAX up-regulation, ROS overproduction and neurite outgrowth impairments. Binding to TRKB, activating its downstream ERK, PI3K-AKT and CREB signaling pathways and reducing caspase activity, LMDS-1/2 exert neuroprotective effects in these cells. Blockage of either ERK or PI3K-AKT signaling pathways counteracted their neuroprotection against tau aggregation, indicating both signaling pathways are indispensable to protect neurons against tau-mediated neurotoxicity. Given that the effects of LMDS-1/2 were only examined in cell models, future studies in animal models of tauopathies or other neurodegenerative diseases will be necessary to validate their potential in treating patients with AD and tauopathies.

## Figures and Tables

**Figure 1 ijms-23-12734-f001:**
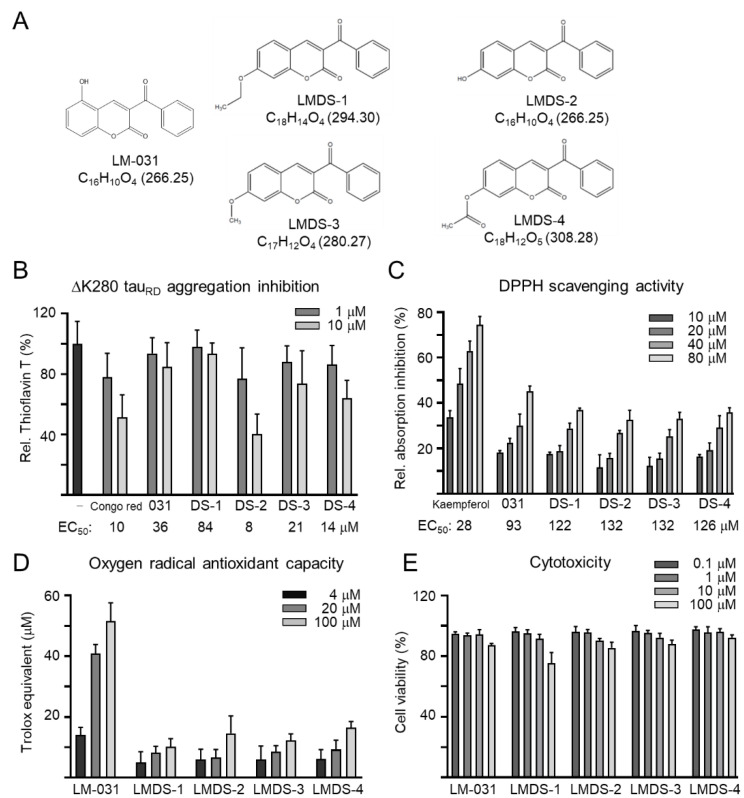
Tested coumarin compounds. (**A**) Structure, formula and molecular weight of LMDS-1 to -4 and LM-031. (**B**) ΔK280 tau_RD_ aggregation inhibition of Congo red (as a positive control), LM-031 and LMDS-1 to -4 (1–10 μM) by the thioflavin T assay (*n* = 3). To normalize, the relative thioflavin T fluorescence of ΔK280 tau_RD_ without compound treatment was set at 100%. Shown below are the EC_50_ values. (**C**) Free radical-scavenging activity of kaempferol (as a positive control), LM-031 and LMDS-1 to -4 (10–80 μM) on DPPH (*n* = 3). Shown below are the EC_50_ values. (**D**) Oxygen radical absorbance capacity of LM-031 and LMDS-1 to -4 (1–100 μM) (*n* = 3). (**E**) Cytotoxicity of LM-031 and LMDS-1 to -4 against ΔK280 tau_RD_-DsRed SH-SY5Y cells examined using the MTT assay. Cells were treated with each test compound (0.1–100 μM) and cell viability was measured the next day (*n* = 3). To normalize, the relative viability of untreated cells was set at 100%.

**Figure 2 ijms-23-12734-f002:**
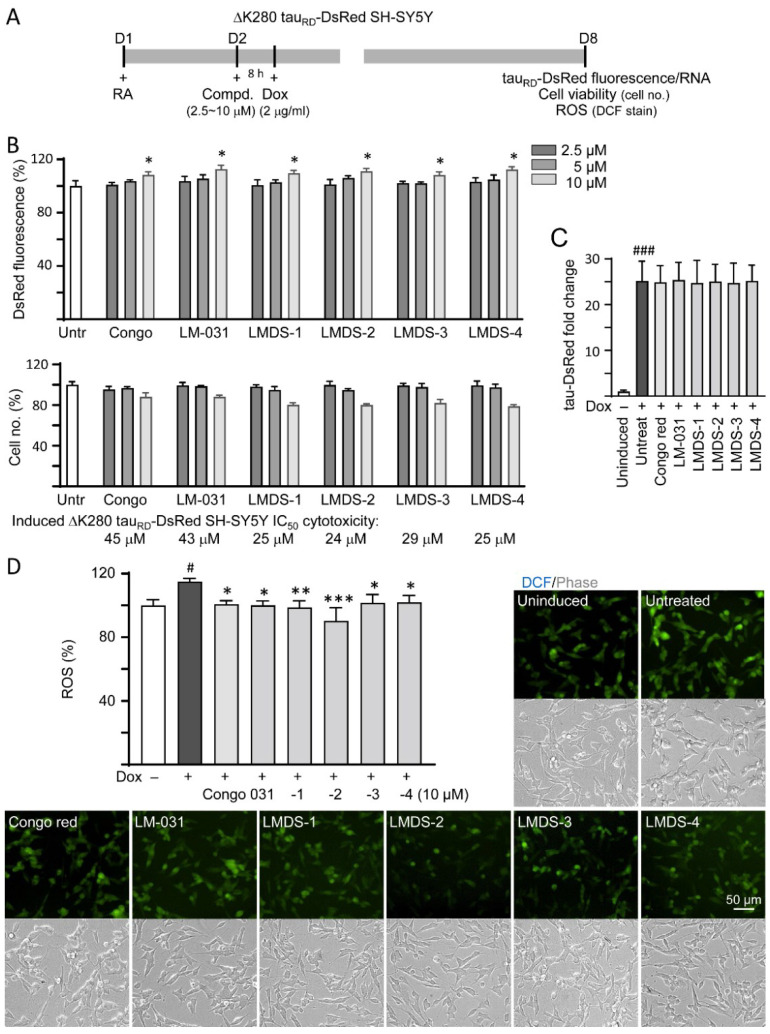
Cellular ΔK280 tau_RD_ aggregation and oxidative stress inhibitory effects of coumarin compounds in ΔK280 tau_RD_-DsRed SH-SY5Y cells. (**A**) Experimental flow chart. On day 1, cells were plated with retinoic acid (RA, 10 µM) added to the culture medium. On day 2, Congo red, LM-031 or LMDS-1 to -4 (2.5–10 µM) was added to the cells for 8 h, followed by inducing ΔK280 tau_RD_-DsRed expression with doxycycline (Dox, 2 µg/mL) for 6 days. On day 8, ΔK280 tau_RD_-DsRed fluorescence, tau_RD_-DsRed RNA and ROS (DCF stain) were measured. (**B**) Assessment of DsRed fluorescence with Congo red, LM-031 or LMDS-1 to -4 (2.5–10 µM) treatment (*n* = 3). Shown below are cell number analyzed in each treatment. The relative DsRed fluorescence/cell number of untreated cells (Untr) was normalized as 100% (two-tailed Student’s *t*-test; *: *p* < 0.05). (**C**) tau_RD_-DsRed RNA of ΔK280 tau_RD_-DsRed cells untreated or treated with Congo red, LM-031 or LMDS-1 to -4 at 10 µM (*n* = 3). HPRT1 was used for normalization. (**D**) Images of DCF stain (green) and ROS assay of ΔK280 tau_RD_-DsRed cells uninduced, untreated or treated with Congo red, LM-031 or LMDS-1 to -4 at 10 µM (*n* = 3). The relative ROS of uninduced cells was normalized (100%). (**C**,**D**) *p* values: comparisons between induced (Dox+) vs. uninduced (Dox−) cells (^#^: *p* < 0.05, ^###^: *p* < 0.001), or compound-treated vs. untreated (induced) (Dox+) cells (*: *p* < 0.05, **: *p* < 0.01, ***: *p* < 0.001) (one-way ANOVA with post hoc Tukey test).

**Figure 3 ijms-23-12734-f003:**
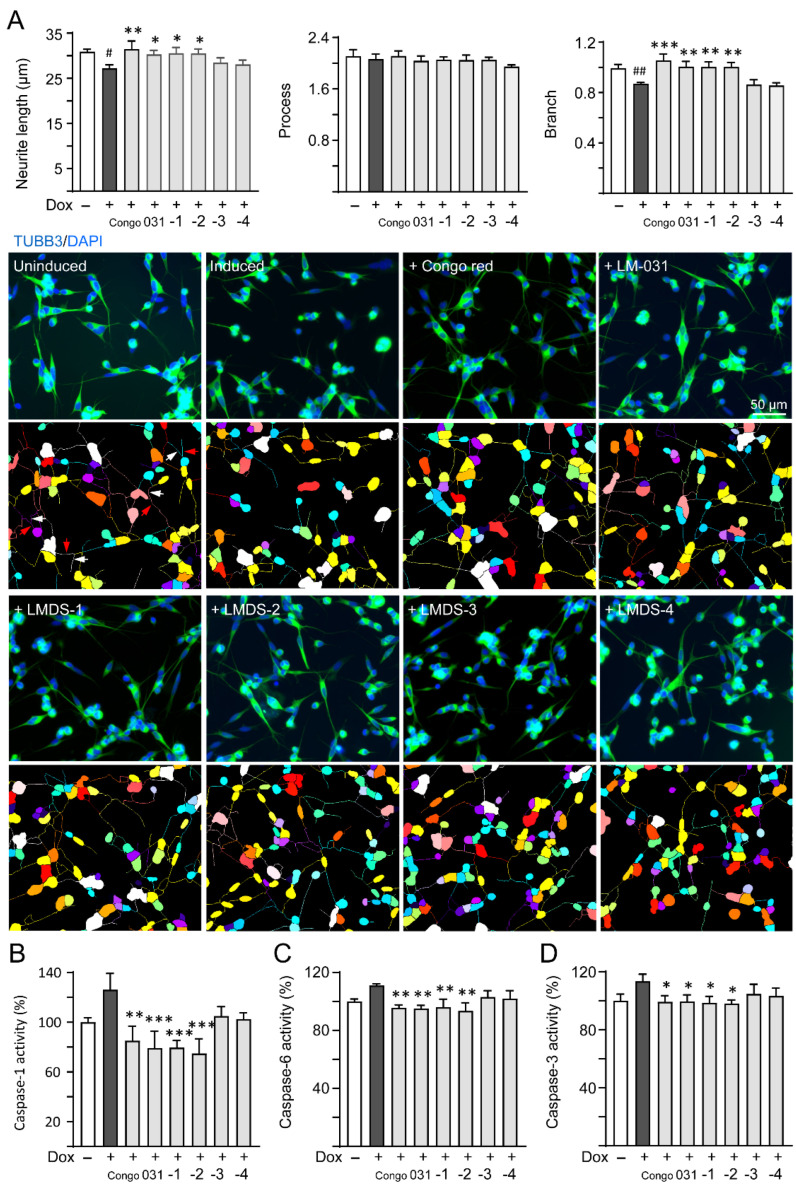
Neuroprotective effects of LM-031 and LMDS-1 to -4 in ΔK280 tau_RD_-DsRed SH-SY5Y cells. (**A**) Neurite outgrowth (length, processes and branching) assay of ΔK280 tau_RD_-DsRed cells uninduced, untreated or treated with Congo red, LM-031 or LMDS-1 to -4 at 10 µM (*n* = 3). Shown below are images of TUBB3 (green)-stained cells with nuclei being counterstained with DAPI (blue), and segmented images with multi-colored mask to assign the outgrowth of a cell body for quantification. In uninduced cells, processes and branches are indicated with red and white arrows, respectively. (**B**) Caspase-1, (**C**) caspase-6 and (**D**) caspase-3 activity assays with Congo red, LM-031 or LMDS-1 to -4 (10 µM) treatment (*n* = 3). The relative caspase-1/caspase-6/caspase-3 activity of uninduced cells (Dox−) was normalized (100%). *p* values: comparisons between induced (Dox+) vs. uninduced (Dox−) cells (^#^: *p* < 0.05, ^##^: *p* < 0.01), or compound-treated vs. untreated (induced) (Dox+) cells (*: *p* < 0.05, **: *p* < 0.01, ***: *p* < 0.001) (one-way ANOVA with a post hoc Tukey test).

**Figure 4 ijms-23-12734-f004:**
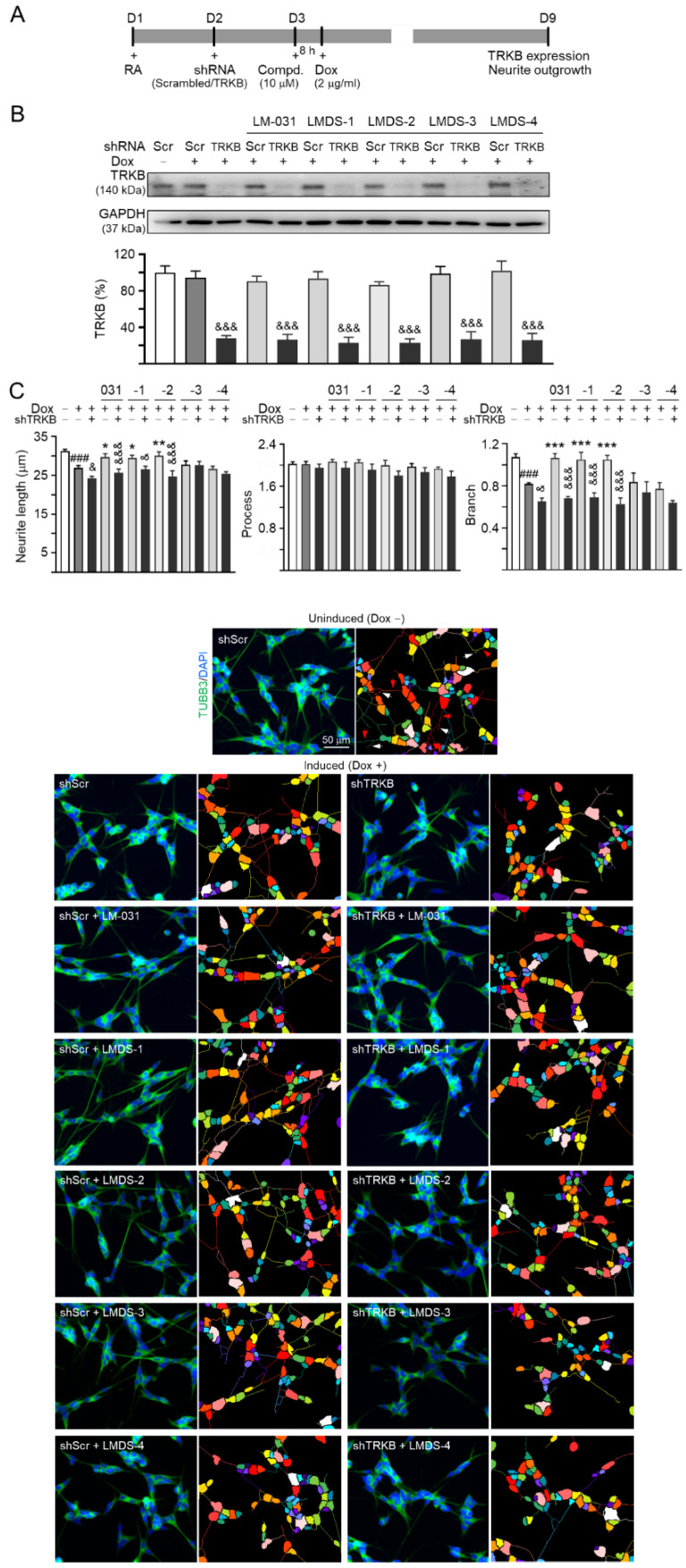
TRKB RNA interference of ∆K280 tau_RD_-DsRed SH-SY5Y cells. (**A**) Experimental flow chart. On day 1, ΔK280 tau_RD_-DsRed SH-SY5Y cells were plated with retinoic acid (RA; 10 µM). On day 2, the cells were infected with lentivirus-expressing TRKB-specific or scrambled shRNA. At 24 h post-infection, LM-031 or LMDS-1 to -4 (10 µM) was added to the cells for 8 h, followed by induction of tau_RD_-DsRed expression (Dox, 2 µg/mL) for 6 days. On day 9, the cells were collected for TRKB and neurite outgrowth analyses. (**B**) Western blot analysis of TRKB in compound-treated cells infected with TRKB-specific or scrambled shRNA-expressing lentivirus (*n* = 3). GAPDH was used as a loading control. To normalize, the relative TRKB of uninduced cells was set at 100%. (**C**) Microscopic images and neurite outgrowth (length, processes and branching) assay of ΔK280 tau_RD_-DsRed-expressing cells with TRKB-specific or scrambled shRNA, and with or without LM-031 or LMDS-1 to -4 (10 μM) treatments (*n* = 3). TUBB3 staining (green) was used to show the extent of neurite outgrowth. Nuclei were counterstained with DAPI (blue). Also shown are segmented images with multi-colored mask being used to assign each outgrowth to a cell body for quantification. In uninduced cells, processes and branches are indicated with red and white arrows, respectively. *p* values: comparisons between induced vs. uninduced cells (^###^: *p* < 0.001), compound-treated vs. untreated (induced) cells (*: *p* < 0.05, **: *p* < 0.01, ***: *p* < 0.001), or TRKB shRNA-treated vs. scrambled shRNA-treated cells (^&^: *p* < 0.05, ^&&&^: *p* < 0.001) (one-way ANOVA with a post hoc Tukey test).

**Figure 5 ijms-23-12734-f005:**
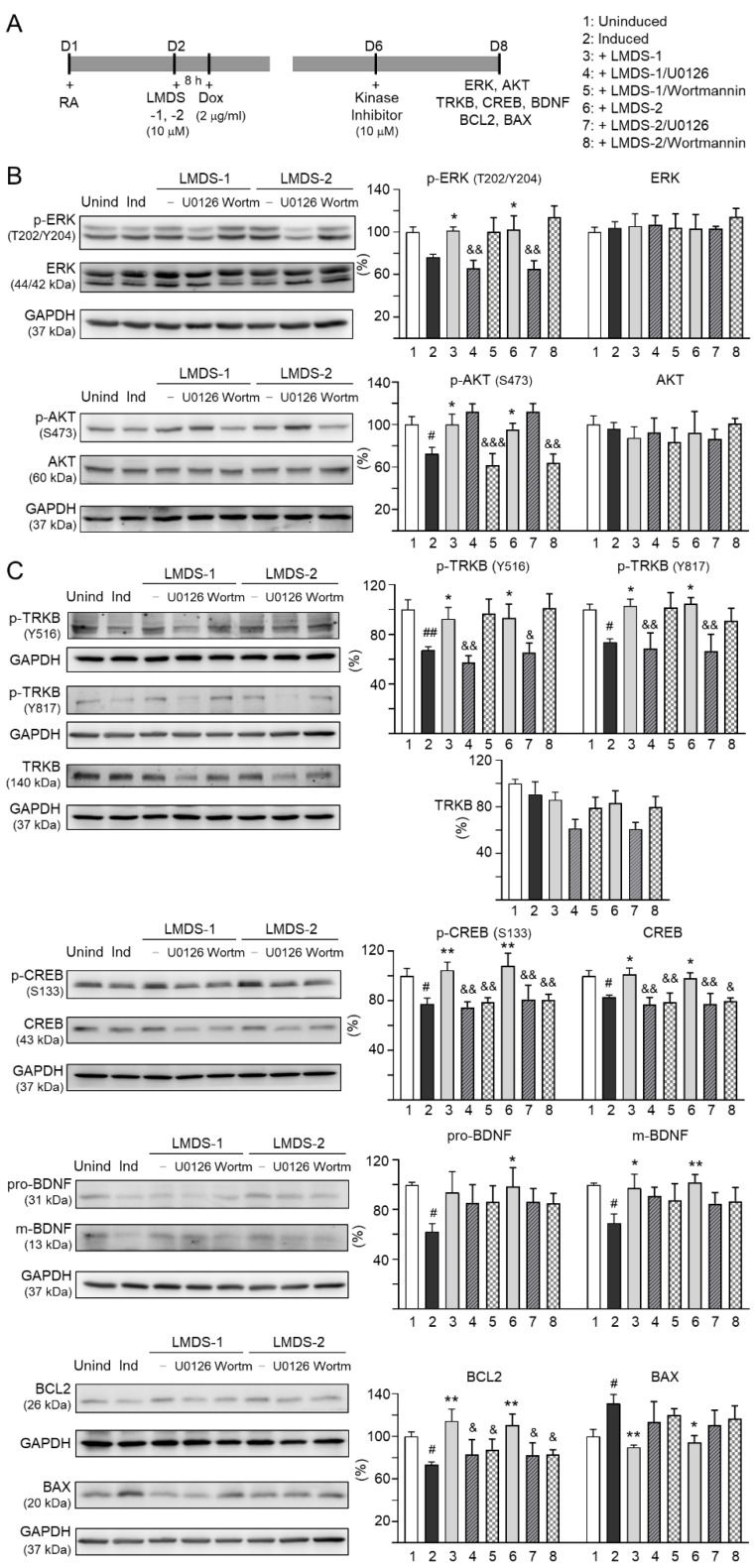
Activation of TRKB signaling in ∆K280 tau_RD_-DsRed SH-SY5Y cells. (**A**) Experimental flow chart. On day 1, cells were plated with retinoic acid (RA, 10 µM) added to the culture medium. On day 2, LMDS-1/2 (10 µM) was added to the cells for 8 h, followed by inducing ∆K280 tau_RD_-DsRed expression with doxycycline (Dox, 2 µg/mL). Kinase inhibitors U0126 or wortmannin (10 µM) were added to the cells on day 6. On day 8, total and/or phosphorylated ERK, AKT, TRKB, CREB, BDNF, BCL2 and BAX levels were measured. (**B**) p-ERK (T202/Y204), ERK, p-AKT (S473), AKT, (**C**) p-TRKB (Y516 and Y817), TRKB, p-CREB (S133), CREB, pro/m-BDNF (31/13 kDa), BCL2 and BAX levels analyzed by immunoblotting using GAPDH as a loading control (*n* = 3). To normalize, protein expression level in untreated cells was set at 100%. *p* values: comparisons between induced vs. uninduced cells (^#^: *p* < 0.05, ^##^: *p* < 0.01), compound-treated vs. untreated cells (*: *p* < 0.05, **: *p* < 0.01), or kinase inhibitor-treated vs. untreated cells (^&^: *p* < 0.05, ^&&^: *p* < 0.01, ^&&&^: *p* < 0.001) (one-way ANOVA with a post hoc Tukey test).

**Figure 6 ijms-23-12734-f006:**
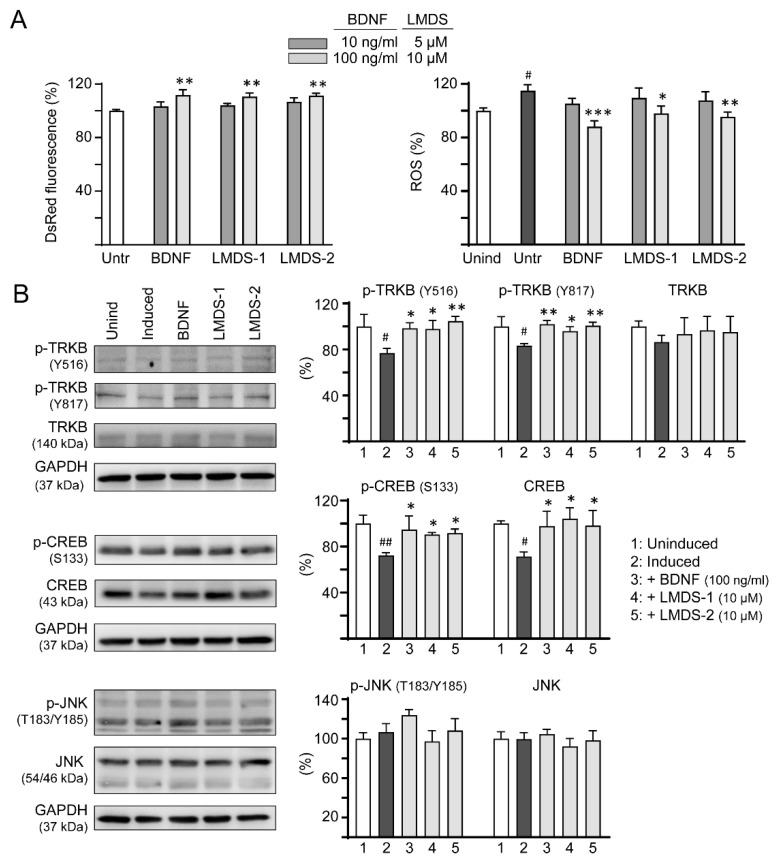
Comparison of BDNF’s and LMDS compound’s signaling activation. (**A**) Assessment of DsRed fluorescence and ROS in ∆K280 tau_RD_-DsRed cells uninduced, untreated, treated with BDNF at 10–100 ng/mL, or LMDS-1, -2 at 5–10 µM (*n* = 3). The relative DsRed fluorescence of untreated cells (Untr) or ROS of uninduced cells (Unind) were normalized (100%). (**B**) Assessment of p-TRKB (Y516/Y817), TRKB, p-CREB (S133), CREB, p-JNK (T183/Y185) and JNK levels with BDNF (100 ng/mL) or LMDS-1, -2 (10 µM) treatment by immunoblot using GAPDH as a loading control (*n* = 3). To normalize, protein expression level in untreated cells was set at 100%. *p* values: comparisons between induced vs. uninduced cells (^#^: *p* < 0.05, ^##^: *p* < 0.01), or compound-treated vs. untreated cells (*: *p* < 0.05, **: *p* < 0.01, ***: *p* < 0.001) (one-way ANOVA with a post hoc Tukey test).

**Figure 7 ijms-23-12734-f007:**
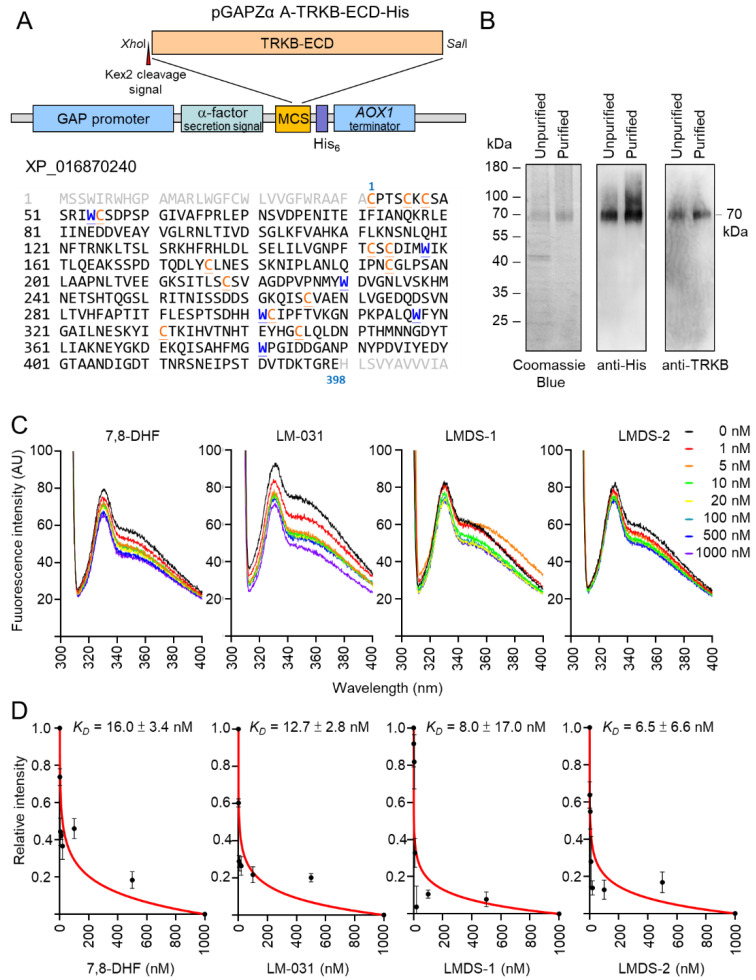
Binding specifically to the extracellular domain of the TRKB receptor. (**A**) pGAPZα A-TRKB-ECD-His plasmid. Human N-terminal extracellular domain of TRKB (TRKB-ECD) was cloned between *Xho*I and *Sal*I sites of multiple cloning site (MCS) of *Pichia pastoris* expression cassette. The TRKB-ECD was in-frame fused to *Saccharomyces cerevisiae* α-factor secretion signal and polyhistidine tag. Expression of the fused gene was driven by *Pichia* glyceraldehyde-3-phosphate dehydrogenase (GAP) promoter. The *AOX1* terminator permits efficient 3’ mRNA processing, including polyadenylation, for increased mRNA stability in *Pichia pastoris*. Shown below are amino acid sequences of TRKB-ECD, with tryptophan (W) marked in blue and cysteine (C) marked in orange. (**B**) The expressed 70 kDa TRKB-ECD protein. The His-tagged TRKB-ECD unpurified and purified from cultured media were examined by Coomassie blue staining of SDS-polyacrylamide gel and immunoblotting using anti-His and anti-TRKB antibodies. (**C**) Tryptophan fluorescence of TRKB-ECD recombinant protein titrated with 7,8-DHF, LM-031, LMDS-1/2. The tryptophan fluorescence of TRKB-ECD excited at 295 nm showed decreased intensity after titrating with different concentrations of the tested compounds. (**D**) The normalized tryptophan intensities at an emission of 330 nm plotted as a function of total added concentrations of the tested compounds. The dissociation constant (*K_D_*) between tested compounds and TRKB-ECD was then calculated by fitting this plot with the equation described.

## Data Availability

All data generated or analyzed during the current study are available from the corresponding author on reasonable request.

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
