# Peer review of "Neuroprotective Action of Coumarin Derivatives through Activation of TRKB-CREB-BDNF Pathway and Reduction of Caspase Activity in Neuronal Cells Expressing Pro-Aggregated Tau Protein"

_ijms, 2022, doi:10.3390/ijms232112734_

Round 1

Reviewer 1 Report

Tau is a neuropathological hallmark of Alzheimer's Disease/tauopathies.

Tau aggregates cause alterations in brain-derived neurotrophic factor (BDNF)/tropomycin receptor kinase B (TRKB)/cAMP-response-element binding protein (CREB) signaling and contribute to neurodegeneration. In this study the authors asked if LM-031, a coumarin derivative with the potential to provide neuroprotection through interaction with TRKB. They report that all four LMDS compounds reduced tau aggregation and reactive oxygen species. Among them, LMDS-1 and -2 reduced caspase-1 and caspase-6 activities and promoted neurite outgrowth, and the effect was significantly reversed by knock-down of TRKB. This is. A nice in vitro study and the results are of interest. However, in vivo work is needed to confirm these promising results. In particular, if the beneficial effects can be extended to:

1)     complement-, and CRP-mediated neuronal death in AD (see, doi: 10.1007/BF00202265; DOI: 10.1038/SREP13281)

2)     improvement of cognitive decline caused by oxidative stress (see, DOI: 10.14336/AD.2015.1022)

Author Response

Reviewer #1:

This is a nice in vitro study and the results are of interest. However, in vivo work is needed to confirm these promising results. In particular, if the beneficial effects can be extended to: 1) complement-, and CRP-mediated neuronal death in AD (see, doi: 10.1007/BF00202265; doi: 10.1038/SREP13281); 2) improvement of cognitive decline caused by oxidative stress (see, doi: 10.14336/AD.2015.1022)

Response: We added sentences and references [51-54] and [63-67] in Discussion to address these issues.

1) lines 367-369: Importantly, ERK signaling can be activated by c-reactive protein or complements [51,52], both of which are colocalized with neurofibrillary tangles in brain of AD patients [52–54].

2) lines 383-385: Importantly, TRKB-PI3K-AKT signaling also promotes the activation of NRF2 antioxidant signaling [63], while disruption of antioxidant signaling increase the production of ROS and further impaired the cognitive function in animal models [64–67].

3) lines 421-422: In vivo experimentation is also needed to confirm these promising results.

Reviewer 2 Report

The manuscript by Te-Hsien Lin et al. presents the activity of newly synthesized compounds that act through TrkB/BDNF system to induce neuronal cell survival and protection against the hyperphosphorylation of mutated tau protein.

Despite the promising results of the 2 new compounds and their significant effects in TrkB activity and neuroprotection in ΔΚ-280 tauRD-DsRed SH-SY5Y cells, important research aspects are poorly explored and the overall significance of this work demands further experimentation. In particular:

1. In all conducted experiments there is no use of BDNF as the most proper positive control. Since the study is focusing on TrkB activation, specific concentrations of BDNF (from 10 to 100ng/ml) should be used to validate the endogenous ligand and the exclusivity of TrkB as mediator of the cell effects. Especially, in the bindings experiments, and in addition to 7,8-DHF use, BDNF (radiolebeled or not) should be tested (indicating also the same or different binding site of the endogenous vs the synthetic compound).

2. Do these synthetic compounds act on p75NTR receptor too as BDNF does? Is this receptor expressed on this cell line?

3. While the binding affinity of LMDS-1/2 is at the range of nM concentration, the cellular effects and the used concentrations of these compounds are at the range of μM. These large concentrations could involve the activation of other receptors too, acting in a pleiotropic manner. Experiments with lower (nM) concentrations of the compounds should be performed.

4. The authors have examined Caspase-1 and -6 but not Caspase-3, which is the most prominent pro-apoptotic mediator in neuronal cells. They also associate amyloid-β deerived effects in these caspases and not tau-related mechanisms of caspase regulation.

5. In the introduction, the authors present findings of tau-mediated regulation of the BDNF/TrkB system. However, this study examines not the downstream effects of tau but the upstream effects of TrkB in tau aggregation, thus, this kind of existing research should be presented.

6. Although the use of a cell line provides significant advantages in order to study cell signaling and effects, a more relative cell system -like primary hippocampal neurons- should be examined. Hippocampal neurons are a well established primary model for Alzheimer Disease pathology and are known to express TrkB receptors and respond to BDNF treatment.

7. Evaluation of cellular effects should be include more assays, like cell death measurement and use of TrkB specific inhibitors (like ANA-12).

Author Response

Reviewer #2:

Despite the promising results of the 2 new compounds and their significant effects in TrkB activity and neuroprotection in ΔΚ-280 tauRD-DsRed SH-SY5Y cells, important research aspects are poorly explored and the overall significance of this work demands further experimentation. In particular:

  1. In all conducted experiments there is no use of BDNF as the most proper positive control. Since the study is focusing on TrkB activation, specific concentrations of BDNF (from 10 to 100ng/ml) should be used to validate the endogenous ligand and the exclusivity of TrkB as mediator of the cell effects. Especially, in the binding experiments, and in addition to 7,8-DHF use, BDNF (radiolebeled or not) should be tested (indicating also the same or different binding site of the endogenous vs the synthetic compound).

Response: (1) We added a new figure (Figure 6, line 279) using BDNF as a positive control to compare the effects of LMDS-1 and -2 on TRKB signaling (lines 257-269): As LMDS-1 and -2 target TRKB and downstream CREB, effects of BDNF and LMDS compounds on TRKB and CREB expression were compared. Inhibition of ΔK280 tauRD aggregation and oxidative stress were first evaluated by treating ∆K280 tauRD-DsRed SH-SY5Y cells with BDNF at 10–100 ng/ml concentration or LMDS-1/2 at 5–10 μM concentration (Figure 6A). BDNF at 100 ng/ml and LMDS-1/2 at 10 μM effectively increased the DsRed fluorescence intensity (111–112%, P = 0.003–0.001) and reduced the ROS induced by ∆K280 tauRD-DsRed overexpression (from 115% to 98–88%, P = 0.012–<0.001). BDNF at 100 ng/ml and LMDS-1/2 at 10 μM were then selected to compare the efficacy in TRKB and CREB expression (Figure 6B). Overexpression of ∆K280 tauRD-DsRed down-regulated p-TRKB (77–83%, P = 0.011–0.010) and p-CREB (72%, P = 0.003), whereas BDNF, LMDS-1 and -2 treatment rescued the reduction (p-TRKB: 96–104%, P = 0.049–0.003; p-CREB: 90–95%, P = 0.044–0.012). No significant differences in p-TRKB and p-CREB were detected between BDNF and LMDS-1/2 groups (P > 0.05).

(2) Mature BDNF has 3 tryptophan residues. BDNF emits intrinsic fluorescence with a maximum at 330 nm when excited at 295 nm. Therefore, binding of BDNF to TRKB-ECD was not included for comparison in this study.

  1. Do these synthetic compounds act on p75NTR receptor (nerve growth factor receptor, NGFR) too as BDNF does? Is this receptor expressed on this cell line?

Response: We added a paragraph and references [35-37] to address these two questions (lines 270-278): In addition to activate TRKB receptor, BDNF also binds to low-affinity nerve growth factor receptor (NGFR) [35], a member of tumor necrosis factor receptor superfamily. NGFR was expressed in the membrane fractions of SH-SY5Y cells [36]. Increased expression of the NGFR in plasma membrane enhanced mitogen activated protein kinase 8 (JNK) activation and apoptotic cell death in SH-SY6Y cells [37]. We thus examined if these LMDS compounds may act on NGFR. As shown in Figure 6B, LMDS-1 and -2 treatments did not increase p-JNK (T183/Y185) expression in SH-SY5Y cells expressing ∆K280 tauRD-DsRed (93–108% versus 107%, P > 0.05), suggesting no enhancement of JNK activation through NGFR binding.

  1. While the binding affinity of LMDS-1/2 is at the range of nM concentration, the cellular effects and the used concentrations of these compounds are at the range of μM. These large concentrations could involve the activation of other receptors too, acting in a pleiotropic manner. Experiments with lower (nM) concentrations of the compounds should be performed.

Response: In ΔK280 tauRD-DsRed cells, treatment with LMDS compounds at nM concentration has no effect on DsRed fluorescence. Therefore, subsequent experiments with lower (nM) concentrations of the compounds were not performed. Regarding the activation of other receptors, we discussed the pleiotropic manner in lines 417-421: Alternatively, LMDS-1/2 may also activate other receptors to modulate above signaling pathways and their neuroprotective effects. Further studies will be warranted to clarify whether ERK, PI3K-AKT, and CREB are upregulated directly by TRKB activation, by aggregate inhibition, or by a pleiotropic mechanism through activation of other receptors.

  1. The authors have examined Caspase-1 and -6 but not Caspase-3, which is the most prominent pro-apoptotic mediator in neuronal cells. They also associate amyloid-β derived effects in these caspases and not tau-related mechanisms of caspase regulation.

Response: We quantified caspase-3 activity (lines 161-162) in Figure 3D (line 163).

  1. In the introduction, the authors present findings of tau-mediated regulation of the BDNF/TrkB system. However, this study examines not the downstream effects of tau but the upstream effects of TrkB in tau aggregation, thus, this kind of existing research should be presented.

Response: We added a sentence and reference [14] in Introduction (lines 52-55): However, in neuronally differentiated SH-SY5Y cells, there is no significant difference in steady state levels of endogenous Tau phosphorylation at Ser202, Thr231, Ser396 and Ser404 between this ΔK280 tauRD-expressed and un-expressed tau pro‐aggregation cell model [14].

  1. Although the use of a cell line provides significant advantages in order to study cell signaling and effects, a more relative cell system -like primary hippocampal neurons- should be examined. Hippocampal neurons are a well-established primary model for Alzheimer disease pathology and are known to express TrkB receptors and respond to BDNF treatment.

Response: We added the related information in Discussion (lines 398-401): Furthermore, our group also demonstrated that LMDS-1 could upregulate TRKB-ERK-CREB pathway and BDNF expression in hippocampal primary neurons incubated with oligomeric Aβ and Aβ-induced AD mice [70].

  1. Evaluation of cellular effects should include more assays, like cell death measurement and use of TrkB specific inhibitors (like ANA-12).

Response: We added IC50 cytotoxicity of test compounds in Figure 2B (lines 117-119 & 126): Based on the cell number analyzed, the IC50 values of congo red, LM-031, LMDS-1, LMDS-2, LMDS-3, and LMDS-4 were 45, 43, 25, 24, 29, and 25 μM, respectively. As TRKB-specific shRNA efficiently reduced TRKB level in ∆K280 tauRD-DsRed cells (only 23-27% left) and neurite outgrowth promoting effect of LMDS-1/2 through TRKB pathway is evident, we did not repeat the experiment with TRKB specific inhibitors.

Reviewer 3 Report

In the present study, the authors investigated the neuroprotective effect of coumarin derivatives against Alzheimer’s disease and other tauopathies, by activation of BDNF-TRKB- 2 CREB signaling and reduction of caspases activities.

Author Response

Reviewer #3:

In the present study, the authors investigated the neuroprotective effect of coumarin derivatives against Alzheimer’s disease and other tauopathies, by activation of BDNF-TRKB- 2 CREB signaling and reduction of caspases activities. In the context of an increasing emergence of these diseases, finding new agents for their treatment (or prevention), is of great interest. Only one minor issue needs to be addressed:

  1. Although in the section “Discussion” the direction of future studies is clearly stated, the limitations of the current study lack.

Response: We added a sentence in Discussion to address this (lines 421-422): In vivo experimentation is also needed to confirm these promising results.

  1. Please take into consideration the manuscript needs moderate English editing (starting with the title).

Response: We carefully edited the manuscript including title (lines 2-4): Neuroprotective Action of Coumarin Derivatives through Activation of TRKB-CREB-BDNF Pathway and Reduction of Caspase Activity in Neuronal cells expressing Pro-Aggregated Tau.

  1. Introduction: You could improve this section by emphasizing the current gap in therapy – explaining in one short sentence why the development of new tau misfolding inhibitors is extremely needed.

Response: We added a sentence and reference [5] in Introduction to address this issue (lines 43-45): In addition, the misfolded tau protein propagates pathology through connected brain circuits in a prion-like manner [5].

Round 2

Reviewer 1 Report

The authors have adequately addressed my concern, the manuscript can be published in its current form

Author Response

Thank you for approving the revised manuscript.

Reviewer 2 Report

The revised manuscript by Lin et al has been significantly improved and it fulfills the requirements for being published in IJMS.

Author Response

(The authors gave the same response as above.)
